# A Fast Loss Model for Cascode GaN-FETs and Real-Time Degradation-Sensitive Control of Solid-State Transformers

**DOI:** 10.3390/s23094395

**Published:** 2023-04-29

**Authors:** Moinul Shahidul Haque, Md Moniruzzaman, Seungdeog Choi, Sangshin Kwak, Ahmed H. Okilly, Jeihoon Baek

**Affiliations:** 1Nexteer Automotive Corp., Saginaw, MI 48601, USA; 2Department of Electrical and Computer Engineering, Mississippi State University, Starkville, MS 39762, USA; 3School of Electrical and Electronics Engineering, Chung-Ang University, Seoul 06974, Republic of Korea; 4Electrical & Electronics and Communication Engineering Department, Koreatech University, Cheonan 31253, Republic of Korea; 5Electrical Engineering Department, Faculty of Engineering, Assiut University, Assiut 71516, Egypt

**Keywords:** cascode GAN-FET, SSTS, switch loss model, degradation aware controller, linear quadratic regulator (LQR), lifetime estimation

## Abstract

This paper proposes a novel, degradation-sensitive, adaptive SST controller for cascode GaN-FETs. Unlike in traditional transformers, a semiconductor switch’s degradation and failure can compromise its robustness and integrity. It is vital to continuously monitor a switch’s health condition to adapt it to mission-critical applications. The current state-of-the-art degradation monitoring methods for power electronics systems are computationally intensive, have limited capacity to accurately identify the severity of degradation, and can be challenging to implement in real time. These methods primarily focus on conducting accelerated life testing (ALT) of individual switches and are not typically implemented for online monitoring. The proposed controller uses accelerated life testing (ALT)-based switch degradation mapping for degradation severity assessment. This controller intelligently derates the SST to (1) ensure robust operation over the SST’s lifetime and (2) achieve the optimal degradation-sensitive function. Additionally, a fast behavioral switch loss model for cascode GaN-FETs is used. This proposed fast model estimates the loss accurately without proprietary switch parasitic information. Finally, the proposed method is experimentally validated using a 5 kW cascode GaN-FET-based SST platform.

## 1. Introduction

In recent years, a number of studies have been conducted to find reliable power electronics alternatives to traditional transformers. Two of the most promising alternatives are solid-state transformers (SSTs) and high-temperature superconducting (HTS) transformers [1,2,3]. While both technologies offer significant advantages over traditional transformers, they differ in their design, capabilities, and potential applications. One of the key advantages of SSTs is their ability to control power flow, which makes them ideal for power electronics applications. SSTs will replace traditional transformers in modern industry, including use in electric vehicle charging and smart-grid applications, due to their low cost, high efficiency (>95%), and compact features [1]. Power semiconductor switches are among the most vulnerable components in a power electronic system (PES) [4]. These switches commonly experience high-frequency (HF) electro-thermal stresses in their SSTs during dynamic operation. Thus, monitoring switch aging and degradation and implementing a proactive degradation-sensitive control strategy are vital for real-world applications [5].

Switches mainly experience HF stresses during the HF dual active bridge (DAB) stage, as shown in Figure 1 [1]. A DAB control strategy minimizes the inductor current to achieve zero-voltage switching (ZVS) [6]. These methods are promising, but they achieve adaptive control without evaluating switch degradation.

The integration of renewable energy sources and new technologies into power systems has led to the development of a new reliability framework for modern power electronic systems [7]. Another study introduced a comprehensive approach that uses a photovoltaic inverter as an example to predict power electronic converter reliability, address wear-out failures, and optimize power system design, operation, and maintenance [8]. With the increasing demand for more electrical systems, reliability assessment and standardization have become increasingly important [7,8]. The development of guidelines is vital to achieving this objective, and current efforts are focused on creating them. Furthermore, this approach is applicable not only to the entire power electronic system but also to individual components. State-of-the-art DAB-control-based protection systems focus on post-fault scenarios [9,10,11,12,13], mainly by carrying out fault tolerance operations after a switch failure. Therefore, they require costly hardware redundancies and high system complexity, which commonly increase the system’s size and weight [9,10]. Redundancy-based network-level power routing is not feasible in standalone SST-based EV charging applications [12]. In these applications, topology transformation methods can bypass fatal switch failure effects for a limited time. However, these methods double the current through the switch, resulting in higher switch stress and accelerated aging, which causes premature SST failure. They also require computationally exhaustive fault detection and isolation procedures. A power-sharing control strategy is proposed to improve the reliability of power converters in DC microgrids by adjusting their loadings based on prior thermal damages, aiming to extend their lifespan and enhance the overall system reliability [14]. Previous studies [12,13,14,15] have relied solely on thermal loading information to determine the stage of degradation. In contrast, accelerated life testing (ALT) can provide additional insights into the degradation pattern, which have not been utilized in this study. In [11] and [13], an adaptive but complicated and costly cooling method for junction temperature control was proposed to extend switch life.

So far, SST control and protection strategies based on monitoring switch degradation have rarely been studied. If successful, these could offer a new, effective solution without costly redundancies. An SST could integrate switch health status into a controller to intelligently identify its health-optimal operating point. Techniques based on thermal cycle counting have a limited capacity to identify switch degradation; instead, they derate the PES based on the number of thermal cycles elapsed [9,10,11,15]. Thus, these methods fail to derate the PES if the switch is degraded early. Switch degradation mapping is critical to utilize the benefits of PES derating.

In this paper, a new degradation-sensitive controller is proposed for a cascode GaN-FET-based SST. The proposed controller uses intelligent thermal-cycle-counting methods and ALT-based switch degradation mapping to ensure safe and robust operation over the SST’s life as well as to enable operation at optimal operating conditions for switch health. A degraded switch deteriorates quickly if the SST continues operating at rated conditions and causes early failure. To address this issue, the proposed controller estimates the optimal switch health condition at which to derate. This intelligent derating allows the SST to reach its target life and avoid unexpected shutdown. A linear quadratic regulator (LQR) determines these rating and derating conditions, estimating the optimal phase shift to ensure the SST’s stability and robustness to system disturbance. A fast-behavioral switch loss model is used to identify switch degradation instantaneously with low complexity, enabling adaptive, real-time operation. This model does not require parasitic parameter estimation and exhaustive computational effort calculation, in contrast to existing analytical loss models. Although our approach is specifically developed for SST, it should be applicable to other power electronics systems as well.

The rest of the paper is organized as follows. In Section 2, the proposed controller’s fundamentals are explained. Online switch lifetime mapping is discussed in Section 3, followed by accelerated-aging-based degradation mapping in Section 4. The design of the proposed controller is presented in Section 5, the design and performance of the LQR in Section 6, and experimental testing and validation in Section 7. The contributions of this paper and future research directions are explored in the conclusion in Section 8.

## 2. Principles of the Proposed Controller

The Coffin–Manson method estimates an application’s remaining useful life based on its operating conditions [16]. This model solely addresses the effect of instantaneous operating conditions on the life of the switch [9,10,11,15], monitoring the number of cycles elapsed to identify the level of degradation. As degradation is random, the switch might degrade at an accelerated rate, so this model includes high variance. Thus, it is important to complement this model with fault-precursor-based identification of degradation severity. Accelerated life testing (ALT)-based degradation mapping provides a switch degradation trajectory, which allows accurate switch-degradation severity assessment and offline planning of degradation-sensitive control [17]. Another essential feature of a degradation-sensitive controller is the ability to identify abrupt changes. The proposed controller intelligently integrates online lifetime-mapping features, degradation mapping based on accelerated life testing, and identification of unexpected degradation to derate the SST for life extension. The principles of the proposed degradation-sensitive controller are shown in Figure 2.

The proposed controller uses information from the following three blocks:i.Online switch lifetime mapping;ii.Accelerated life testing (ALT)-based degradation mapping;iii.Identification of unexpected degradation.

In block i, the number of cycles to failure (N_f_) is estimated online based on the proposed behavioral switch loss model. In block ii, switch degradation is mapped based on the switch’s fault precursor trajectory under ALT. On-state resistance (R_DS,ON_) shows the highest sensitivity to switch degradation in cascode GaN-FETs [17,18]. This R_DS,ON_ trajectory is statistically analyzed, and a degradation probability is mapped based on ALT. In block iii, R_DS,ON_ is measured online and regularly evaluated to identify any sudden changes in switch health.

The proposed controller has two functions—(i) a supervisory function and (ii) an operational function, as shown in Figure 2. In the supervisory function, dynamic programming is used to identify the switch’s health status and estimate the optimal degradation-sensitive operating conditions based on the three blocks of inputs. In the operational function, an LQR regulates the optimal phase-shift angle based on the set degradation-sensitive operating point.

The following sections briefly describe the functions of the three blocks of the proposed controller.

## 3. Online Switch Lifetime Mapping

Block i estimates N_f_ based on the junction temperature, which is estimated using a switch loss model. The procedure is shown in Figure 3. This block’s three main components are the switch loss model, R-C Foster-network-based junction temperature estimation, and lifetime estimation. The switch loss causes are the mean junction temperature (T_J,m_) and junction temperature variation (ΔT_J_). Wire-bond lift-off and solder fatigue are the two dominant open-circuit failure mechanisms triggered by HF thermo-mechanical stress. A mismatched coefficient of thermal expansion (CTE) between different layers increases the severity of this stress [17,19]. These dominant mechanisms result in wire-bond cracks and solder degradation or eventual lift-off. In the following subsections, these three components are discussed.

### 3.1. Switch Loss Model for Cascode GaN-FETs

A fast and accurate behavioral loss model is proposed. In the cascode structure, a low-voltage (LV) Si-MOSFET is cascoded with a normally-on GaN-HEMT. The turn-off loss is different, especially from IGBT, due to the absence of a tailing current. An analytical loss model for cascode GaN-FETs was developed in [20,21,22,23,24] based on complex cascode-structure-induced parasitic capacitances and inductances. This analytical model thus requires a computationally exhaustive process and proprietary information. Additionally, this model requires expensive testing for parameter extraction. Piecewise linear models are faster but less accurate, and the effect of the PCB parasitic is not considered [25]. The proposed model was developed by analyzing and modeling the switching transition behavior of drain-source voltage (V_DS_) and drain current (I_d_) in the half-bridge configuration to overcome these challenges.

Moreover, this behavioral model uses parameters easily extractable from the datasheet and double pulse test (DPT). This behavioral model reflects the effects of parasitic capacitances and commutation inductors. Thus, the proposed model is adaptable to high-frequency applications.

The switch loss has two components: switching loss (P_sw_) and conduction loss (P_cond_), calculated as follows:(1)$$P_{\mathrm{Loss}}=P_{\mathrm{cond}}+P_{\mathrm{sw}}$$

Conduction loss can be estimated as follows:(2)$$P_{\mathrm{cond}}=I_{L}^{2}R_{\mathrm{DS},\mathrm{ON}}\;$$
where I_L_ is the RMS load current and R_DS,ON_ is the function of temperature and switch health status. P_sw_ is the summation of turn-on and turn-off power losses.

#### 3.1.1. Turn-on Loss Calculation

Commonly, a cascode GaN-FET’s turn-on is modeled by LV Si-MOSFET, normally on GaN-HEMT’s complex physics-based interactions. The proposed behavior-based model inherently addresses the effects of parasitic capacitors and commutation inductors and is thus accurate. The proposed model does not require the solving of complex high-degree polynomials and differential equations. The switching transition of V_DS_ and I_d_ during turn-on is shown in Figure 4a, and is divided into four regions, as follows:

**Region I:** LV Si Gate Charging

The LV Si-MOSFET controls the switching transitions of the cascode GaN-FET. The turn-on process starts when the LV Si-MOSFET’s gate voltage reaches the threshold and a conducting channel is established in LV Si-MOSFET. The gate-drive loss (P_dri_) can be expressed as follows:(3)$$P_{\mathrm{dri}}\left(t\right)=Q_{G}V_{G}f_{s}$$
where Q_G_ is the gate charge, V_G_ is the gate voltage, and f_s_ is the switching frequency. I_d_ starts rising when the gate-source voltage (V_GS_) of GaN-HEMT reaches its threshold and causes turn-on loss in cascode GaN-FET due to V-I overlapping. This V-I overlapping starts at t_1_, as shown in Figure 4a.

**Region II:** Increasing Drain Current

In this region, the GaN-HEMT is fully turned on. I_d_ increases linearly and reaches load current (I_L_). Thus, I_d_ can be modeled in the proposed model as follows:(4)$$I_{d}\left(t\right)=t\frac{d}{\mathrm{dt}}I_{\mathrm{rise}}$$
where $0\leq I_{d}<I_{L}$ and $t_{1}\leq t<t_{2}$, and dI_rise_/d_t_ is the rising rate of I_d_, which is a constant. The value of dIrise/dt depends on the commutation inductances. The value is estimated from the DPT. During this period, V_DS_ is assumed to be constant at V_DS,OFF_. The energy loss during this period is expressed as follows:(5)$$E_{\mathrm{turn}-\mathrm{on}-\mathrm{II}}=\int_{t_{0}}^{t_{1}}V_{\mathrm{DS}}\left(t\right)I_{d}\left(t\right)\mathrm{dt}=\frac{1}{2}{\left(t_{1}-t_{o}\right)}^{2}V_{\mathrm{DS},\mathrm{OFF}}\left(\frac{d}{\mathrm{dt}}I_{\mathrm{rise}}\right)$$

**Region III:** Decreasing drain-source voltage

In the half-bridge, two switches in one leg switch complementarily. In region III, I_L_ supplies I_d_ to the top switch and reverse-conducting current (I_RR_) to the bottom switch. I_d_ rises at dI_rise_/d_t_ until it reaches I_L_ + I_RR_. There are two sub-regions in region III. In sub-region a, I_d_ increases due to commutation inductances and internal parasitics. The behavior of I_d_ is modeled as follows:(6)$$I_{d}\left(t\right)=t\frac{d}{\mathrm{dt}}I_{\mathrm{rise}}+I_{L}$$
where $I_{L}\leq I_{d}<I_{L}+I_{\mathrm{RR}}$ and $t_{2}\leq t<t_{22}$. In this sub-region, dIrise/dt remains the same as in region II. V_DS_ linearly drops to an off-state V_DS_ (V_DS,OFF_) during this time. This drop is due to the increase in I_d_ over I_L_. The behavioral model of V_DS_ can be expressed as follows:(7)$$V_{\mathrm{ds}}\left(t\right)=V_{\mathrm{ds},\mathrm{off}}+t\frac{d}{\mathrm{dt}}V_{\mathrm{DS},1}$$
where dV_DS,1_/d_t_ is the falling rate of V_DS_. The energy loss in this sub-region is as follows:(8)$$\begin{array}{cc} E_{turn-on-III} & =\int_{t_{2}}^{t_{22}}V_{DS}(t)I_{d}(t)dt\\ & =\frac{1}{3}(t_{22}-t_{2}{)}_{3}\left(\frac{d}{dt}V_{DS}{}_{,1}\right)\left(\frac{d}{dt}I_{rise}\right)+\frac{1}{2}{\left(t_{22}-t_{2}\right)}^{2}V_{DS,off}\\ & \left(\frac{d}{dt}I_{rise}\right)+\frac{1}{2}{\left(t_{22}-t_{2}\right)}^{2}I_{L}\left(\frac{d}{dt}V_{DS}{}_{,1}\right)+\left(t_{22}-t_{2}\right)V_{DS,off}I_{L} \end{array}$$

In subregion b, the reverse recovery charge needs to be removed from the bottom switch. The rate of change in I_d_ is defined by the reverse recovery current of the bottom switch. In this sub-region, I_d_ falls from I_L_ + I_RR_ to I_L_ when V_DS_ falls from V_DS,OFF_,1 to V_DS,ON_. The internal parasitics of the cascode structure cause these transitions. The analytical model requires internal parasitic information, which is not available. In the proposed model, the behaviors of I_d_ and V_DS_ can be modeled as follows:(9)$$I_{d}\left(t\right)=\left(I_{L}+I_{\mathrm{RR}}\right)-t\frac{d}{\mathrm{dt}}I_{\mathrm{fall}}$$
(10)$$V_{\mathrm{DS}}\left(t\right)=V_{\mathrm{DS},\mathrm{Off},1}-t\frac{d}{\mathrm{dt}}V_{\mathrm{DS},\mathrm{fall}}$$

This model inherently addresses the effects of parasitics, which affect the rates of change in V_DS_ and I_d_. Thus, the modeling approach ensures speed and accuracy. The energy loss is calculated as follows:(11)$$\begin{array}{cc} E_{\mathrm{turn}-\mathrm{on}-\mathrm{III},\mathrm{fall}}= & \int_{t_{22}}^{t_{3}}V_{\mathrm{DS}}\left(t\right)I_{d}\left(t\right)\mathrm{dt}\\ & =\left(I_{L}+I_{\mathrm{RR}}\right)V_{\mathrm{DS},\mathrm{Off},1}\left(t_{3}-t_{22}\right)-\frac{1}{2}{\left(t_{3}-t_{22}\right)}^{2}\left(\frac{d}{\mathrm{dt}}I_{\mathrm{fall}}\right)V_{\mathrm{DS},\mathrm{Off},1}-\frac{1}{2}(I_{L}\\ & +I_{\mathrm{RR}}){\left(t_{3}-t_{22}\right)}^{2}\left(\frac{d}{\mathrm{dt}}V_{\mathrm{DS},\mathrm{fall}}\right)+\frac{1}{3}{\left(t_{3}-t_{22}\right)}^{3}\left(\frac{d}{\mathrm{dt}}I_{\mathrm{fall}}\right)\left(\frac{d}{\mathrm{dt}}V_{\mathrm{DS},\mathrm{fall}}\right) \end{array}$$

**Region IV:** Ringing Region

In this region, I_d_ and V_DS_ behave like a damping system and reach their steady states at I_L_ and V_DS,ON_, respectively. These tendencies can be modeled as follows:(12)$$I_{d}\left(t\right)=A_{1}\exp(-\alpha_{1}t)\sin(\omega_{01}t+\theta )+I_{L}\;$$
(13)$$V_{\mathrm{DS}}\left(t\right)=A_{2}\exp(-\alpha_{1}t)\sin(\omega_{01}t+\psi )+V_{\mathrm{DS},\mathrm{ON}}$$
where A_1_ and A_2_ are the amplitudes of I_d_ and V_DS_, respectively; α_1_ is the decay rate of I_d_ and V_DS_; and ω_01_ is the frequency of I_d_ and V_DS_, respectively. At the start of this region, I_d_ is I_L_, V_DS_ is V_DS,ON_, and θ and ψ are zero. The loss in this region is as follows:(14)$$E_{\mathrm{turn}-\mathrm{on}-\mathrm{IV}}=\int_{t_{3}}^{t_{4}}V_{\mathrm{DS}}\left(t\right)I_{d}\left(t\right)\mathrm{dt}\;$$
where E_turn-on-IV_ is a function of ω_01_, α_01_, A_1_, and A_2_. These parameters are estimated from the double-pulse test. The total turn-on loss is the summation of the losses in regions I, II, and III:(15)$$E_{\mathrm{on}}=E_{\mathrm{turn}-\mathrm{on},\mathrm{II}}+E_{\mathrm{turn}-\mathrm{on},\mathrm{III}}+E_{\mathrm{Turn}-\mathrm{on},\mathrm{IV}}$$

#### 3.1.2. Turn-off Loss Calculation

Like turn-on loss, turn-off loss is modeled by analyzing the switching transition behavior. This model is fast and easily implementable without any proprietary information and costly testing. The turn-off transition of V_DS_ and I_d_ during the turn-off process is shown in Figure 4b, where the turn-off region is divided into three regions.

**Region-I:** Gate Capacitor Discharge

The turn-off process starts when V_g_ is zero and initiates the discharge process of the gate-source capacitor of the LV Si-MOSFET. In this region, there are insignificant changes in V_DS_ and I_d_. Thus, there is only an insignificant gate-drive loss, similar to that in Equation (3).

**Region II:** Decreasing Drain Current

In region II, GaN-HEMT is shut down due to the interaction between the gate capacitance of the LV MOSFET and GaN-HEMT. I_d_ decreases from I_L_ to zero. V_DS_ increases from V_DS,ON_ to V_ds2,off_, which is greater than the input voltage (V_in_). These behaviors of I_d_ and V_DS_ can be modeled as follows:(16)$$I_{d}\left(t\right)=I_{L}+t\frac{d}{\mathrm{dt}}I_{\mathrm{fall}}\;$$
where I_d_(t_1_) = I_L_ at t = t_1_ and I_d_(t_2_) = 0 at t = t_2_.
(17)$$V_{\mathrm{DS}}\left(t\right)=V_{\mathrm{DS},\mathrm{on}}+t\frac{d}{\mathrm{dt}}V_{\mathrm{DS},\mathrm{rise}}$$
where V_DS_(t_1_) = V_DS,ON_ at t = t_1_ and VDS(t_2_) = V_DS,off_ at t = t_2_. These behavioral models address the effects of the parasitic components on the switching transition while avoiding complex modeling and exhaustive calculations. The energy loss in this region is as follows:(18)$$E_{\mathrm{turn}-\mathrm{off}-\mathrm{II}}=\int_{t_{1}}^{t_{o}}V_{\mathrm{DS}}\left(t\right)I_{d}\left(t\right)\mathrm{dt}=I_{L}V_{\mathrm{DS},\mathrm{on}}\left(t_{1}-t_{o}\right)-\frac{1}{2}{\left(t_{1}-t_{o}\right)}^{2}\left(\frac{d}{\mathrm{dt}}I_{\mathrm{fall}}\right)\;-\frac{1}{2}I_{L}{\left(t_{1}-t_{o}\right)}^{2}\left(\frac{{\mathrm{dV}}_{\mathrm{DS},\mathrm{rise}}}{\mathrm{dt}}\right)+\frac{1}{3}{\left(t_{1}-t_{o}\right)}^{3}\left(\frac{{\mathrm{dV}}_{\mathrm{DS},\mathrm{rise}}}{\mathrm{dt}}\right)\left(\frac{d}{\mathrm{dt}}I_{\mathrm{fall}}\right)\;$$

**Region III:** Ringing Region

In region III, I_d_ and V_DS_ reach steady-state conditions at zero and V_DS,OFF_, respectively. During this period, I_d_ and V_DS_ behave like an underdamped system. This behavior is modeled as follows:(19)$$I_{d}\left(t\right)=A_{3}\exp(-\alpha_{2}t)\sin(\omega_{02}t)$$
(20)$$V_{\mathrm{DS}}\left(t\right)={\Delta V}_{\mathrm{DS}}\exp(-\alpha_{2}t)\cos(\omega_{o2}t)$$
where, at t = t_3_, I_d_ = 0 and V_DS_ = V_DS,OFF_, A_3_ is the amplitude of the overdamped system, α_2_ is the decaying rate, and ω_o2_ is the decaying frequency. The turn-off loss is as follows:(21)$$E_{\mathrm{turn}-\mathrm{off}-\mathrm{III}}=\int_{t_{2}}^{t_{3}}V_{\mathrm{DS}}\left(t\right)I_{d}\left(t\right)\mathrm{dt}$$
where E_turn-on-III_ is a function of ω_02_, α_02_, A_3_, and ΔV_DS_. These parameters are estimated from the DPT. The total turn-off loss is the summation of the losses as follows:(22)$$E_{\mathrm{off}}=E_{\mathrm{turn}-\mathrm{off},\mathrm{II}}+E_{\mathrm{turn}-\mathrm{off},\mathrm{III}}$$

This behavioral model is a fast and efficient alternative to complex analytical loss models. Addressing parasitic capacitances and commutation inductances using the time series behavioral model results in fast and accurate switch loss estimation.

### 3.2. R-C Foster-Model-Based Junction Temperature Estimation

The R-C Foster model is the second component in block I, as shown in Figure 3. Switch loss is translated into T_J_ using the R-C Foster model as follows:(23)$$T_{J}={\;P}_{\mathrm{loss}}Z_{\mathrm{th}\;}+{\;T}_{c}$$
where T_J_ is the switch’s junction temperature, Z_th_ is the thermal impedance, and T_C_ is the case temperature. Z_th_ is estimated as follows:(24)$$Z_{\mathrm{th}}=\overset{n}{\underset{i=1}{\sum }}r_{i}(1-e^{\frac{-t}{\tau_{i}}})$$
where r_i_ is the thermal resistance of the switch and τi is the thermal time constant. Thermal time constraints are expressed as τ_i_ = r_i_ C_th,i_, where C_th,i_ is the thermal capacitance. These thermal parameters are estimated from the switch’s transient thermal impedance curve provided in the datasheet.

### 3.3. Lifetime Estimation

The Coffin–Manson life estimation model relates Nf to T_J,m_ and ΔT_J_. This model estimates the instantaneous variation in N_f_ due to a change in operating conditions. The model can be expressed as follows:(25)$$N_{f}=\;A\left({\Delta T}_{J}^{-b_{1}}\right)\exp\left(\;\frac{b_{2}}{T_{j,\;m}+273}\right)\;$$
where T_J,m_ is the minimum junction temperature and A, b_1_, b_2_, and b_3_ are the empirical coefficients. These empirical coefficients and the estimated value, T_J_, include estimation error uncertainties. Switch lifetime consumption accelerates due to changes in ΔT_J_ and T_J,m_, as follows:(26)$$D.A.F={\left(\frac{{\Delta T}_{j,1}}{{\Delta T}_{j,2}}\right)}^{-b1}\exp\left(\;\frac{b_{2}}{T_{j_{1},\;m}+273}-\frac{b_{2}}{T_{j_{2},\;m}+273}\right)$$
where D.A.F is the degradation acceleration factor when the operation point changes from (ΔT_J1_, T_J1,m_) to (ΔT_J2_, T_J2,m_) due to switch degradation. The damage to the switch is estimated based on Miner’s linear damage rule as follows:(27)$$C=\sum \frac{n_{i}}{N_{f}}$$
where C is the switch’s consumed lifetime and n_i_ is the number of cycles consumed by the switch. The information on consumed lifetime is used with ALT-based switch degradation mapping to address deviations in the operating conditions and switch lifetime acceleration.

## 4. Degradation Mapping Based on Accelerated Life Testing

The typical lifetime of a power semiconductor switch is 10–12 years [5]. ALT maps this trajectory to a logical timeframe. ALT of a cascode GaN-FET provides critical insight into the relationship between R_DS,ON_ trajectory dynamics and consumed life. The R_DS,ON_ trajectory shows sensitivity to packaging-related failures in the cascode GaN-FET, such as wire-bond lift-off (WBLO) and solder crack [18,26]. Switch degradation is mapped by analyzing the R_DS,ON_ trajectories.

In this paper, a power-cycling ALT is used for mapping the R_DS,ON_ trajectory. The ALT conditions and switch degradation mappings are provided in Section 7.2. The R_DS,ON_ trajectories show that there are three distinct regions in the life of the cascode GaN-FET—(i) the healthy region, (ii) the slow degradation (SD) or constant degradation (CD) region, and (iii) the exponential degradation (ED) region. These tendencies in the R_DS,ON_ trajectory are shown in Figure 5a. There are also R_DS,ON_ trajectories that are outliers to the typical trajectories, as shown in Figure 5b. These outlier trajectories bias the mean R_DS,ON_ trajectory. It is logical to model the degradation trajectory using the median R_DS,ON_ trajectory, which is not significantly affected by outliers. This median trajectory is modeled as follows:(28)$$P_{t}\left(R_{\mathrm{DS},\mathrm{ON}}\leq R_{\mathrm{DS},\mathrm{ON},\mathrm{med}}\right)=\frac{1}{2}$$
where R_DS,ON, med_ is the median of the R_DS,ON_ trajectories. This mapping of degradation using Equation (28) is shown in Figure 5b.

## 5. A Degradation-Sensitive Controller for SSTS

The proposed controller operates using two functions: a supervisory function and an operational function, as shown in Figure 2. The dynamically programmed supervisory function determines a switch-health-sensitive operating point. Based on this operating point, the LQR process estimates the optimal phase shift for the SST.

### 5.1. Dynamic Programmed Supervisory Function

Dynamic programming uses switch degradation mapping to estimate the necessary operating conditions to achieve a target lifetime for the SST. To achieve the target lifetime under the proposed method, the SST operates using the optimal derating trajectory based on the cost function, as follows:(29)$$J=\min\left(C_{\mathrm{rated}}-\sum_{i=1}^{3}C_{\deg\mathrm{raded},}{}_{i}\right)$$
where C_rated_ is the rated lifetime under constant junction temperature variation and C_degraded,i_ is the consumed lifetime. The cost function is minimal when the rated value is equal to a switch’s lifetime. The objective of the dynamically programmed supervisory function is to maximize J by adaptively selecting operating conditions based on switch health status. This strategy is shown graphically in Figure 6. This strategy ensures constant junction temperature variation and uniform lifetime consumption. The controller is designed as follows:

Minimize: ${\mathrm{Arg}}_{\max}C_{\deg\mathrm{raded}}=\left(C_{1}+C_{2}+C_{3}\right)$

Subject to:

Constraint 1: $C_{1}>0$, $C_{2}>0$, $C_{3}>0$

Constraint 2: $0.8P_{\mathrm{rated}}\leq P\leq P_{\mathrm{rated}}$

Constraint 3: $C_{\deg\mathrm{radate}}\leq C_{\mathrm{rated}}$

Constraint 4: $P_{\mathrm{II}}>P_{\mathrm{III}}$

where C_1_, C_2_, and C_3_ are consumed life in regions I, II, and III, respectively. The derating algorithm is shown in Figure 6. As the switch will be more degraded in the exponential degradation (ED) region than in the slow degradation (SD) region, it is logical to impose higher derating in the ED region. Based on the constraints, the allowed operating conditions are shown in Figure 7. When the switch is healthy, the SST operates as rated. However, the SD and ED regions’ operating conditions are programmed based on constraints 1–4. If the SST operates at P_min_, it will have a maximum lifetime, but it will violate constraint 3. If the SST operates at P_rated_, J will be maximized, but the SST will fail before its target lifetime. Different combinations of operating conditions in these two regions lead to different lifetimes for the SST. To integrate these dynamic operating conditions, Equation (26) is used to estimate lifetime consumption under each condition. E_on_ and E_off_ are integrated into the proposed controller as a look-up table to reduce the computational burden.

### 5.2. LQR-Based Operational Function

A regulator is required to control the DAB stage in SST to deliver the reference voltage. In the proposed controller, the operational function uses an LQR as this regulator. In this sub-section, this LQR is described; the LQR ensures the optimal phase-shift angle for the rated or derated operating condition set by the supervisory function. The LQR shows robust performance under system disturbance and estimates the optimal phase-shift angle to ensure efficiency. The LQR design requires a state-space model of the DAB stage. The equivalent circuit in the DAB is shown in Figure 8, where L is the HF transformer’s leakage inductance, VAB is the inverter output, and VCD is the rectifier input voltage.

The average model of the output stage of the DAB is shown in Figure 8. Using Kirchhoff’s current law,
(30)$$\mathrm{Co}\frac{\mathrm{dVo}}{\mathrm{dt}}+\frac{\mathrm{Vo}}{R_{L}}-\frac{\mathrm{ViD}\left(1-D\right)}{2{\mathrm{Lf}}_{s}}=0$$
where V_i_ is the input voltage, V_o_ is the output voltage, RL is the load, C_o_ is the output capacitor, and f_s_ is the switching frequency. The small-signal model of this output average model for small variations in v_o_ and d is as follows:(31)$$\frac{{\mathrm{dv}}_{o}}{\mathrm{dt}}=\frac{-\mathrm{vo}}{R_{L}C_{o}}+\frac{V_{i}\left(1+2D\right)}{2{\mathrm{LC}}_{o}f_{s}}d$$

The state-space model of the DAB is as follows:(32)$$\overset{˙}{\underset{}{x}}=\mathrm{Ax}+\mathrm{Bu}$$
where
$$A=\left[\begin{array}{cc} 0 & 1\\ 0 & -\frac{1}{R_{L}C_{o}} \end{array}\right],\;B=\left[\begin{array}{c} 0\\ \frac{V_{i}\left(1+2D\right)}{2{\mathrm{LC}}_{o}f_{s}} \end{array}\right]x=\left[\begin{array}{c} \int v_{o}\mathrm{dt}\\ v_{o} \end{array}\right]\mathrm{and}\;u=d$$

In this state-space model, a new variable, ʃv_o_dt, is introduced, which results in zero steady-state error. The quadratic cost function of the system is as follows:(33)$$J_{\mathrm{LQR}}=\int \left(x^{T}\mathrm{Qx}+u^{T}\mathrm{Ru}\right)\mathrm{dt}$$
where J_LQR_ is the quadratic cost function, Q is a 2-by-2 positive semi-definite matrix, and R is a scalar that should be positive. The closed-loop poles’ locations depend on the choices of Q and R.

In this paper, R = 1 and Q = [q_1_ 0; 0 q_2_]. Thus, q_1_ and q_2_ determine the speed and damping of the system. For the state feedback, it is assumed that
(34)u=−kx
where $k=\left[\begin{array}{cc} k_{1} & k_{2} \end{array}\right]$ represents the feedback gain.

Putting Equation (34) into Equation (35), it is found that
(35)$$J_{\mathrm{LQR}}=\frac{1}{2}\int \left[x^{T}\left(Q+k^{T}\mathrm{Rk}\right)x\right]\mathrm{dt}$$

The optimal solution for k is found as follows:(36)$$k=R^{-1}B^{T}P$$
where P is the solution of the algebraic Recatti equation, as follows:(37)$$\mathrm{PA}+A^{T}P-{\mathrm{PBR}}^{-1}B^{T}+Q=0$$

## 6. LQR Design and Its Performance Analysis

The LQR design requires the operating point information as shown in Equation (30). The operating frequency of the SST is 50 kHz; the rated voltage and power are 400 V and 5 kW, respectively; the leakage inductance is 53 μH; and the output capacitance is 120 μF. Based on this operating condition and the system information, the LQR is designed to have sufficient controller speed and zero steady-state error. The root locus and the step response of the modeled LQR controller are shown in Figure 9a,b with R = 1, Q = [50 0;0 10^(−5.5)], and K = [−7.0711 0.0048]. The settling time of the system is 0.043 s, and the overshoot is less than 5%. Thus, the LQR is operating with reasonable speed and accuracy.

## 7. Experimental Testing and Validation

### 7.1. Validation of the Behavioural Switch Loss Model

Commutation inductance due to the PCB layout plays a vital role in switching loss estimation. These inductances change with the PCB layout, which is application dependent. Thus, it is essential to conduct a DPT on the same PCB layout as the SST to address these inductances’ effect on switching loss. V_DS_ and Id measurements require high-bandwidth probes. A differential voltage probe (TMDP0200) and coaxial shunt 0.1 Ω SSDN-10 current sensor are used for fast, high-precision measurements. V_DS_ and I_d_ are shown in Figure 10 during the turn-on and turn-off transitions. Turn-on and turn-off losses are estimated by calculating energy losses due to the crossover of V_DS_ and I_d_. Moreover, these signals are properly aligned to improve the integrity of the testing.

The rates of change in V_DS_ and I_d_ during turn-on and turn-off are estimated from the DPT test and the datasheet, along with ringing region parameters. These parameters are used in Equations (3)–(25) for switching loss estimation and are integrated into the controller. The estimated switching loss is compared with the measured loss from the DPT at different operating conditions to validate the proposed switching loss model, as shown in Figure 11. This behavioral switching-loss model closely follows the experimental switching loss tendency and has a 2.3% average mean squared error. The model is fast, parameter extraction is easy, and the model follows the experimental model’s loss tendency closely.

### 7.2. Validation of the Degradation-Sensitive Controller

An experimental setup for the degradation-sensitive controller is shown in Figure 12. The operating frequency is 50 kHz, the rated input and output voltage is 400 V, the rated power is 5 kW, and the leakage inductance is 53 μH.

The R_DS,ON_ trajectories of the cascode GaN-FET under ALT are shown in Figure 13. In the power cycling ALT test, the switch’s case temperature was varied between 25 °C and 100 °C using active-switch heating. To measure V_DS,ON_ and monitor R_DS,ON_ in real time, a signal-conditioning circuit was utilized. For a detailed description of this circuit, please refer to our previous publication [19], and the shunt resistor in the phase leg was used to measure I_d_. The R_DS,ON_ measurement was sampled at a steady state to avoid the effect of switching transients. To reduce the effect of noise, 50 R_DS,ON_ samples were averaged. These R_DS,ON_ samples were also temperature scaled.

It was observed that, until 60% of the way through their life, the switches were in the healthy region. In this region, ΔR_DS,ON_ is between 0% and 2%. From 60% to 80% of the switches’ lifetime, the switches were in the SD region, where ΔR_DS,ON_ is between 2% and 7%. When ΔR_DS,ON_ is greater than 7%, the switch is in the ED region. The statistically modeled median R_DS,ON_ and the dynamically programmed estimated optimal operating trajectory are shown in Figure 14.

The SST was operated with a new switch, a 40% degraded switch, and an 80% degraded switch to validate the proposed control system’s applicability. The ΔR_DS,ON_ values were 1.6% and 7% for the 40% and 80% degraded switches, respectively. The switches were degraded using ALT. The SST operates at a rated inductor current of V_o_ = 400 V when the switch is healthy, as shown in Figure 15. As a 40% degraded switch is also in the healthy region, the SST keeps operating in the rated condition. The inductor current is decreased when the degradation-sensitive degradation controller identifies the switch as being in the ED region, as shown in Figure 15. The optimal operating condition was identified as V_o_ = 360 V.

The derated operating condition reduces the junction temperature experienced by the switch. The lifetime profiles for different values of T_J,m_ and ΔT_J_ are shown in Figure 16. The PDFs and cumulative density functions (CDFs) of N_f_ at (ΔT_J_ = 10 °C, T_J,m_ = 50 °C) and (ΔT_J_ = 11 °C, T_J,m_ = 51 °C) are shown in Figure 17. When T_J,m_ and ΔT_J_ are reduced by 1 °C, the CDF becomes less steep. If the switch keeps operating in the rated condition, it will fail at 96% of the rated lifetime. The proposed derating condition considers switch degradation and achieves the rated lifetime, extending the switch’s life by 4%. This extended life is crucial for scheduling maintenance before the switch fails.

## 8. Conclusions

This paper proposes a degradation-sensitive controller for an SST that intelligently derates the system’s power based on the switch’s health in order to prevent system failure before the end of the switch’s expected lifetime. The proposed controller increases the SST’s lifetime by 4% over a traditional controller by derating the SST to 80% of its rated power. The proposed approach involves mapping switch degradation to derating levels, with the system operating at its rated conditions until 60% of its consumed life, followed by a gradual reduction of power levels based on switch degradation, and a forced shutdown if degradation exceeds 90%, effectively extending the lifetime of the system. This lifetime extension is achieved by reducing T_J,m_ and ΔT_J_ limits by 1 °C. The objective of this strategy is to maintain a consistent temperature variation while ensuring that T_J,m_ does not increase. This is particularly important because an increase in its value can significantly reduce the device’s lifetime. Although the proposed controller derates the SST with switch degradation, this extended lifetime is significant for maintenance scheduling and avoidance of unexpected failure.

The proposed fast behavioral cascode GaN-FET switch loss model shows an average mean squared error of 2.3% and does not require proprietary switch information. The model parameters are easily extractable from the datasheet and DPT, and the loss model does not require exhaustive computation. Though it is developed for SST, it should be applicable to other power electronics systems as well. The integration of health-monitoring systems and degradation-sensitive controllers into digital twin and cloud computing platforms could be useful for industrial and electric transport maintenance and asset management.

## Figures and Tables

**Figure 1 sensors-23-04395-f001:**
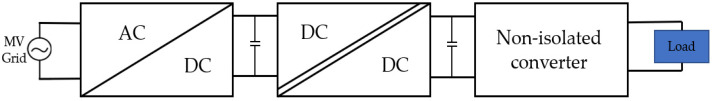
Block diagram of the solid-state transformer (SST) for an EV charging application.

**Figure 2 sensors-23-04395-f002:**
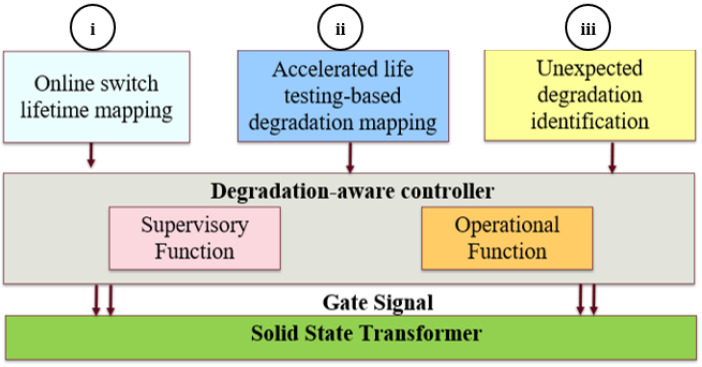
Structure of the proposed degradation-sensitive SST controller.

**Figure 3 sensors-23-04395-f003:**
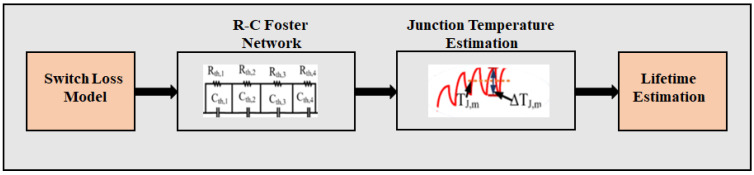
Block structure of the online switch lifetime mapping.

**Figure 4 sensors-23-04395-f004:**
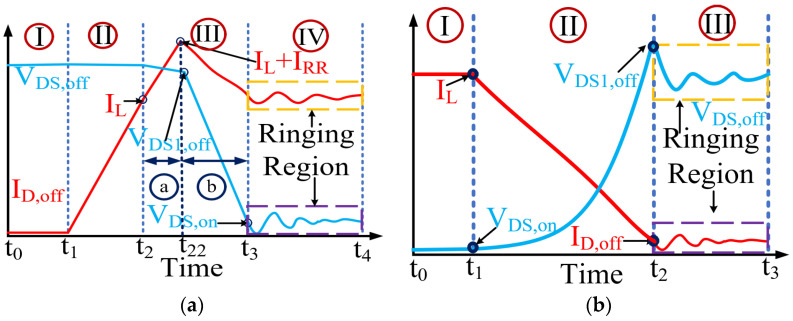
V_DS_ and I_d_ of a cascode GaN-FET: (**a**) during turn-on transition; (**b**) during turn-off transition.

**Figure 5 sensors-23-04395-f005:**
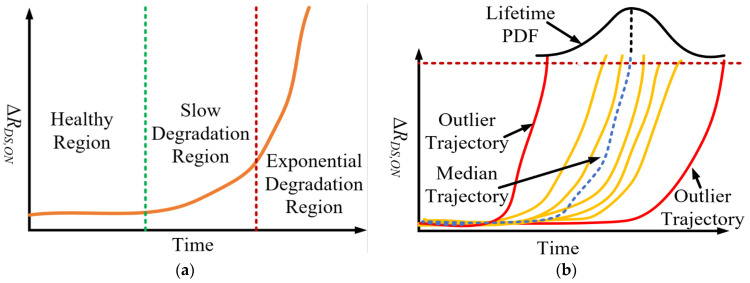
(**a**) R_DS,ON_ trajectory for a cascode GaN-FET; (**b**) median R_DS,ON_ trajectory for a cascode GaN-FET.

**Figure 6 sensors-23-04395-f006:**
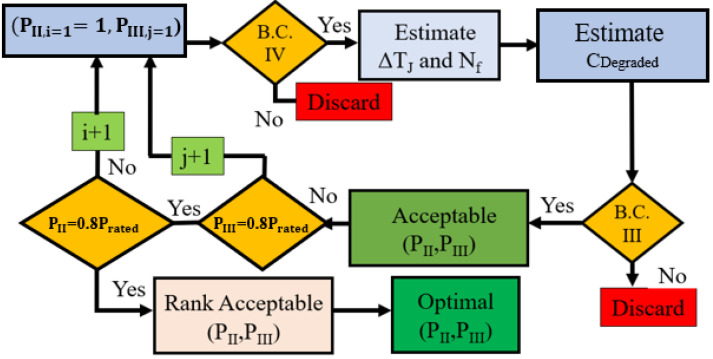
Optimal operating condition estimation algorithm.

**Figure 7 sensors-23-04395-f007:**
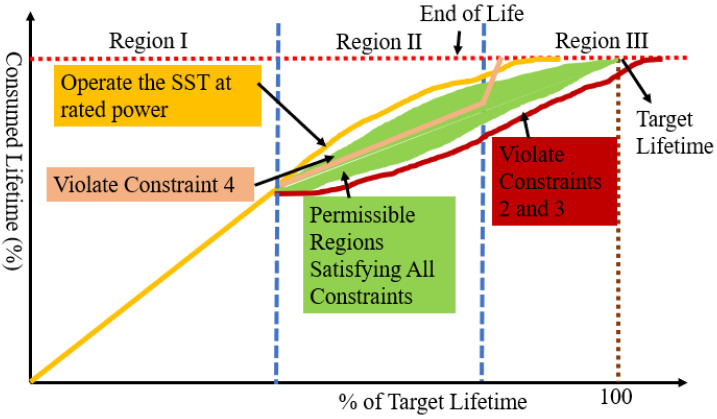
Degradation-sensitive dynamically programmed operating conditions.

**Figure 8 sensors-23-04395-f008:**
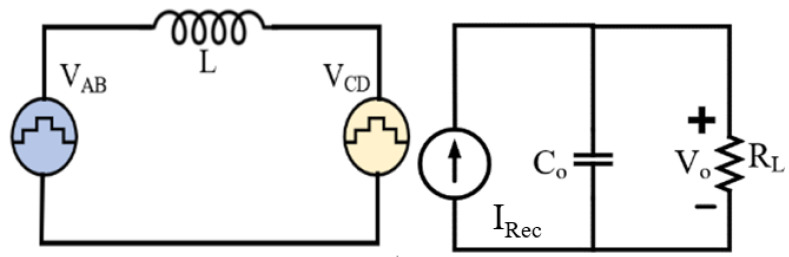
Equivalent circuit of the DAB stage of the SST.

**Figure 9 sensors-23-04395-f009:**
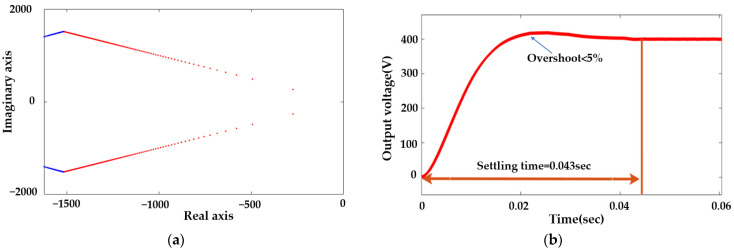
Performance of the designed LQR controller: (**a**) root locus; (**b**) step response.

**Figure 10 sensors-23-04395-f010:**
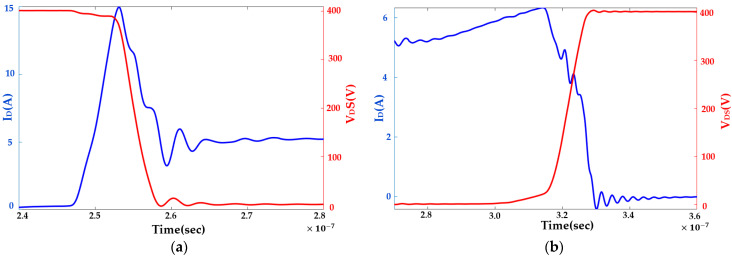
Drain-source voltage and drain current: (**a**) during turn-on; (**b**) during turn-off.

**Figure 11 sensors-23-04395-f011:**
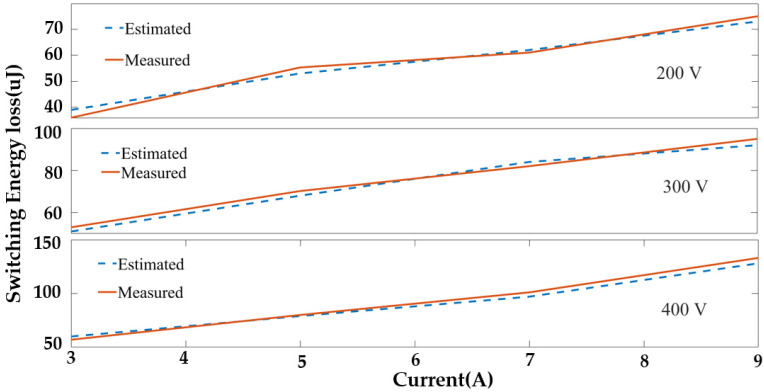
Estimated and measured switching energy loss under different operating conditions.

**Figure 12 sensors-23-04395-f012:**
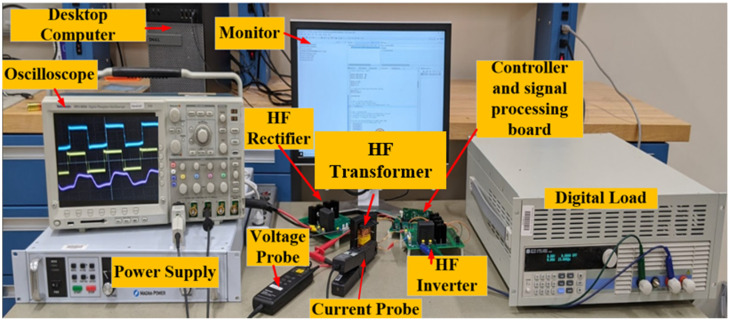
Experimental setup for a cascode GaN-FET-based SST.

**Figure 13 sensors-23-04395-f013:**
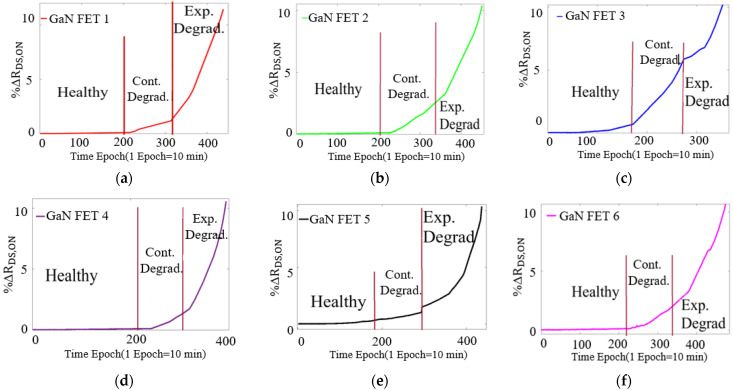
Actual trajectory of R_DS,ON_: (**a**) GaN-FET1, (**b**) GaN-FET2, (**c**) GaN-FET3, (**d**) GaN-FET4, (**e**) GaN-FET5, and (**f**) GaN-FET6.

**Figure 14 sensors-23-04395-f014:**
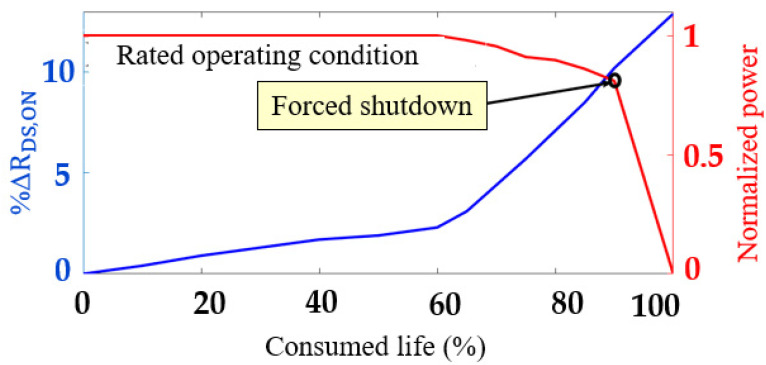
R_DS,ON_ trajectory mapping and dynamically programmed operating point mapping.

**Figure 15 sensors-23-04395-f015:**
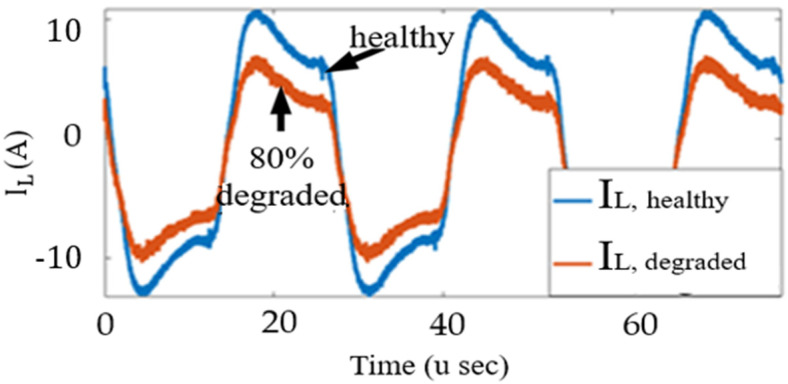
Inductor current in the rated condition and in a degraded condition.

**Figure 16 sensors-23-04395-f016:**
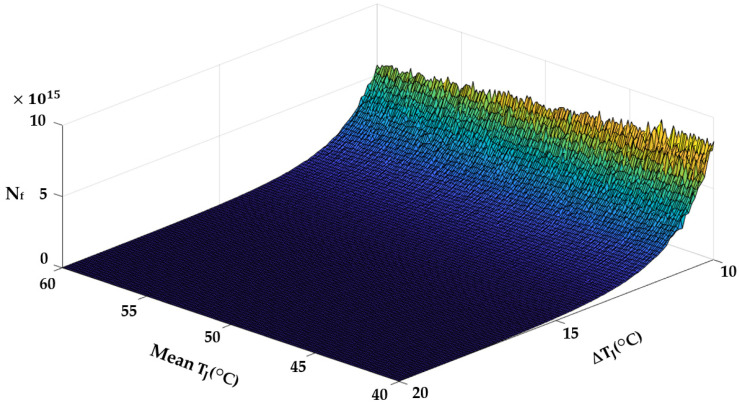
Lifetime profiles for different values of ΔT_J_ and T_J,m_.

**Figure 17 sensors-23-04395-f017:**
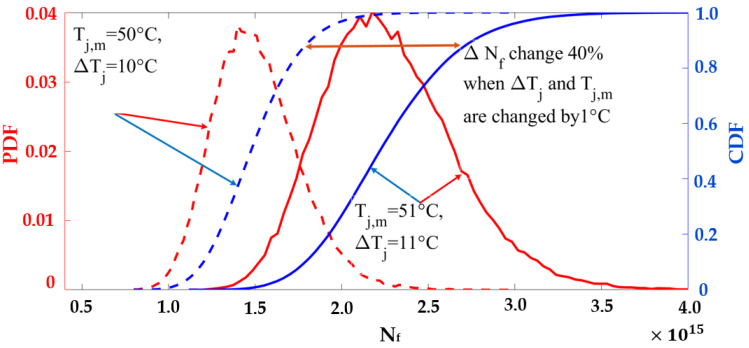
The effects of ΔT_J_ and T_J,m_ on switch lifetime and switch degradation.

## Data Availability

Not applicable.

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
