# Peer review of "A Fast Loss Model for Cascode GaN-FETs and Real-Time Degradation-Sensitive Control of Solid-State Transformers"

_sensors, 2023, doi:10.3390/s23094395_

Round 1

Reviewer 1 Report

Fig. 1. is not like a solid state transformer and is like to a serial connection of rectifier and dc-dc converter. why?

what are the differences between performance of a conventional transformer and a solid-state transformer. rated frequency, voltage, current, tap-changing, etc.

please give references for this sentence "SSTs will replace traditional transformers in modern industry, including in electric 31 vehicle charging and smart-grid applications, due to their low cost, high efficiency (> 32 99%), and compact features".

please compare solid-state transformers with other modern technologies of transformers like high-temperature superconducting transformers.

you can see below two survey papers in this regard:

F. Irannezhad and H. Heydari, "Conducting a Survey of Research on High Temperature Superconducting Transformers," in IEEE Transactions on Applied Superconductivity, vol. 30, no. 6, pp. 1-13, Sept. 2020, Art no. 5500613, doi: 10.1109/TASC.2020.2983970.

Moradnouri, A., Ardeshiri, A., Vakilian, M. et al. Survey on High-Temperature Superconducting Transformer Windings Design. J Supercond Nov Magn 33, 2581–2599 (2020). https://doi.org/10.1007/s10948-020-05539-6

 Minor editing of English language required.

Author Response

Review Report 1. sensors-2346577

  • Fig. 1. is not like a solid-state transformer and is like to a serial connection of rectifier and dc-dc converter. why?

à Answer:

Thank you for your valuable opinion on the most important part of the paper.

This paper presents a novel degradation-sensitive controller designed for a cascode GaN-FET-based solid-state transformer (SST). In the Introduction (Page 2, lines 81-82), we have included a block diagram of the solid-state transformer (SST) for an EV charging application, which illustrates the HF dual active bridge (DAB) stage of switches. This structure is commonly used in solid-state transformers (SSTs). To provide further clarity, we have also included a reference https://doi.org/10.1002/2050-7038.12996 to support this explanation.

Contents before change (Introduction Page 1 line 39)

Switches mainly experience HF stresses during the HF dual active bridge (DAB) stage, as shown in Fig. 1.

Contents after change (Introduction Page 2 line 46-47 )

Switches mainly experience HF stresses during the HF dual active bridge (DAB) stage, as shown in Fig. 1 [1].

  • What are the differences between performance of a conventional transformer and a solid-state transformer. rated frequency, voltage, current, tap-changing, etc.

à Answer:

Thank you for providing valuable feedback regarding the performance differences between conventional transformers and solid-state transformers.

Solid-state transformers (SSTs) differ from conventional transformers in several ways. Conventional transformers normally operate at 50 or 60 Hz and have a limited voltage and current rating. In contrast, SSTs can operate at much higher frequencies like few kHz and can provide higher voltage and current levels. Solid-state transformers (SSTs) can regulate voltage without requiring a tap changer. Another key difference is that the power electronic converter stages of the SST enable full-range control of the terminal voltages and currents, which is not possible with conventional low-frequency (LF) transformers (LFTs). This provides greater flexibility and control over the operation of the SST, making it a more versatile and efficient alternative to conventional transformers.

  • Please give references for this sentence "SSTs will replace traditional transformers in modern industry, including in electric vehicle charging and smart-grid applications, due to their low cost, high efficiency (> 99%), and compact features".

Thank you for providing valuable feedback regarding the need for references about solid-state transformers (SSTs) efficiency.

We apologize for the mistake in our initial statement in Introduction (Page 1, lines 31-33) regarding the high efficiency of solid-state transformers (SSTs). We have revised our statement to reflect the accurate efficiency value of higher than 95% and have included a reference https://doi.org/10.1002/2050-7038.12996 to support this claim. We hope that this revision will address any confusion caused by our earlier statement and provide accurate information to our readers.

Contents before change (Introduction Page 1 line 31-33)

SSTs will replace traditional transformers in modern industry, including in electric vehicle charging and smart-grid applications, due to their low cost, high efficiency (> 99%), and compact features.

Contents after change (Introduction Page 1 line 38-40)

SSTs will replace traditional transformers in modern industry, including in electric vehicle charging and smart-grid applications, due to their low cost, high efficiency (> 95%), and compact features [1].

  • Please compare solid-state transformers with other modern technologies of transformers like high-temperature

superconducting transformers.

à Answer:

Thank you for your valuable feedback.

Solid-state transformers (SSTs) are power electronic devices that use semiconductor components to convert and regulate electrical energy. SSTs have gained popularity in recent years due to their ability to offer greater control and flexibility in power electronics applications. Due to our emphasis on power electronics applications, we have chosen to exclusively use solid-state transformers to meet our specific requirements. Although other alternatives are available, solid-state transformers offer the best solution for our needs.

Added content

In recent years, a number of studies have been conducted to find the reliable power electronics alternatives to traditional transformers. Two of the most promising alternatives are solid-state transformers (SSTs) and high-temperature superconducting (HTS) transformers. While both technologies offer significant advantages over traditional transformers, they differ in their design, capabilities, and potential applications. One of the key advantages of SSTs is their ability to control power flow, which makes them ideal for power electronics applications.

(Page 1 line 32-38).

  • You can see below two survey papers in this regard:

à Answer:

Thank you for your valuable suggestion to refer to the two survey papers https://ieeexplore.ieee.org/document/9054959 and https://doi.org/10.1007/s10948-020-05539-6. We greatly appreciate your feedback.

We incorporated these two references regarding High-Temperature Superconducting Transformer into introduction of our manuscript, where we discuss reliable power electronics alternatives to traditional transformers.

Contents before change (Introduction Page 1 line 32-34)

Two of the most promising alternatives are solid-state transformers (SSTs) and high-temperature superconducting (HTS) transformers.

Contents after change (Introduction Page 1 line 32-35)

Two of the most promising alternatives are solid-state transformers (SSTs) and high-temperature superconducting (HTS) transformers [1-3].

Reviewer 2 Report

In this paper, a novel, degradation-sensitive, adaptive SST controller for cascode GaN-FETs is proposed. The proposed controller uses accelerated life testing (ALT)-based switch degradation mapping for degradation severity  assessment. This controller intelligently derates the SST to 1) ensure robust operation over the SST’s  lifetime and 2) achieve the optimal degradation-sensitive function. A fast-behavioral switch-loss model for cascode GaN-FETs is also proposed.

1. The abstract should be narrow down on the problem and highlight the need of the proposed work with experimental results.

2. Add the contents in the abstract of the paper and highlight the impact of the proposed work. Result and discussion should be rewritten to summarize the findings/significance of the work. 

3. To explore Comparative results with existing approaches/methods relating to the proposed work. The method/approach in the context of the proposed work should be written in detail. 

4. At Line 296, what is WBLO?

5. In Figure 5.(a), how to determine three stages?

6. The inspiration of your work must be highlighted. For example, http://dx.doi.org/10.1109/TCSS.2022.3152091;https://doi.org/10.1111/iwj.13723;https://doi.org/10.1016/j.ins.2022.12.068;http://dx.doi.org/10.1016/j.oceaneng.2022.113424 and so on.

The authors are requested to correct all spelling mistakes.

Author Response

  1. The abstract should be narrow down on the problem and highlight the need of the proposed work with experimental results.

à Answer:

Thank you for your feedback on the abstract. We have revised the abstract (Abstract Page 1 line 14-26) to narrow down on the problem and highlight the need for the proposed work with experimental results. Specifically, we have emphasized the limitations of the current state-of-the-art degradation monitoring methods for power electronics systems and how the proposed degradation-sensitive, adaptive SST controller addresses these challenges. We have also provided information on the proposed accelerated life testing (ALT)-based switch degradation mapping and fast-behavioral switch-loss model, as well as the experimental validation of the proposed method using a 5kW cascode GaN-FET-based SST platform. We hope these changes provide a clearer understanding of the problem, proposed solution, and experimental results in the paper.

Contents before change (Abstract Page 1 line 14-26)

Solid-state transformers (SSTs) are promising solutions in electric vehicle charging, smart homes, and smart grid applications. Unlike in traditional transformers, a semiconductor switch’s degradation and failure can compromise its robustness and integrity. It is vital to continuously monitor a switch's health condition to adapt it to mission-critical applications. State-of-the-art degradation monitoring methods are computationally intensive, have a limited capacity to identify the severity of degradation, and are challenging to implement in real time. In this paper, a novel, degradation-sensitive, adaptive SST controller for cascode GaN-FETs is proposed. The proposed controller uses accelerated life testing (ALT)-based switch degradation mapping for degradation severity assessment. This controller intelligently derates the SST to 1) ensure robust operation over the SST’s lifetime and 2) achieve the optimal degradation-sensitive function. A fast-behavioral switch-loss model for cascode GaN-FETs is also proposed. This proposed fast model estimates the loss accurately, without proprietary switch parasitic information. The proposed method is validated through a 5kW cascode GaN-FET-based SST platform.

Contents after change (Abstract Page 1 line 14-27)

This paper proposes a novel, degradation-sensitive, adaptive SST controller for cascode GaN-FETs. Unlike in traditional transformers, a semiconductor switch’s degradation and failure can compromise its robustness and integrity. It is vital to continuously monitor a switch's health condition to adapt it to mission-critical applications. The current state-of-the-art degradation monitoring methods for power electronics systems are computationally intensive, have limited capacity to accurately identify the severity of degradation, and can be challenging to implement in real-time. These methods primarily focus on conducting accelerated life testing (ALT) of individual switches and are not typically implemented for online monitoring. The proposed controller uses accelerated life testing (ALT)-based switch degradation mapping for degradation severity assessment. This controller intelligently derates the SST to 1) ensure robust operation over the SST’s lifetime and 2) achieve the optimal degradation-sensitive function. Additionally, a fast-behavioral switch-loss model for cascode GaN-FETs is used. This proposed fast model estimates the loss accurately, without proprietary switch parasitic information. Finally, the proposed method is experimentally validated using a 5kW cascode GaN-FET-based SST platform.

  1. Add the contents in the abstract of the paper and highlight the impact of the proposed work. Result and discussion should be rewritten to summarize the findings/significance of the work.

à Answer:

Thank you for your valuable feedback on our paper. We have made the necessary revisions to address your concerns.

Specifically, we have added more details to the abstract to better highlight the impact of our proposed work mentioned in previous answer. Additionally, we have rewritten the mentioned sections to more effectively summarize the significance of our findings.

Once again, we appreciate your constructive feedback and hope that our revisions have adequately addressed your concerns. Please let us know if you have any further suggestions or questions.

Contents before change (Introduction Page 2 line 72-74)

This model does not require parasitic parameter estimation and exhaustive computational effort calculation, in contrast to existing analytical loss models.

Contents after change (Introduction Page 2 line 95-98)

This model does not require parasitic parameter estimation and exhaustive computational effort calculation, in contrast to existing analytical loss models. Although our approach was specifically developed for SST, it should be applicable to other power electronics system as well.

Contents before change (Conclusion Page 15 line 472-486)

In this paper, a degradation-sensitive controller for an SST has been proposed to intelligently derate the system’s power by considering the switch’s health, in order to prevent system failure before the end of the switch’s expected lifetime. This degradation-sensitive controller provides a 4% increase in life over a traditional controller by derating the SST to 80% of its rated power. This lifetime extension is achieved by reducing TJ, m and ΔTJ limits by 1oC. Although the proposed controller derates the SST with switch degradation, this extended lifetime is significant for maintenance scheduling and avoidance of unexpected failure.

The proposed fast behavioral cascode GaN-FET switch-loss model shows a 2.3% average mean-squared error. This model does not require proprietary information about the switch. The parameters are also easily extractable from the datasheet and DPT. This loss model does not require exhaustive computation. Integrated into digital twin and cloud computing platforms, these health-monitoring systems and degradation-sensitive controllers will be useful for industrial and electric-transport maintenance and asset management.

Contents after change (Conclusion Page 15 line 498-519)

This paper proposes a degradation-sensitive controller for an SST that intelligently derates the system's power based on switch’s health, in order to prevent system failure before the end of the switch's expected lifetime. The proposed controller increases the SST's lifetime by 4% over a traditional controller by derating the SST to 80% of its rated power. The proposed approach involves mapping switch degradation to derating levels, with the system operating at its rated conditions until 60% of its consumed life, followed by a gradual reduction of power levels based on switch degradation, and a forced shutdown if degradation exceeds 90%, effectively extending the lifetime of the system. This lifetime extension is achieved by reducing TJ, m and ΔTJ limits by 1°C. The objective of this strategy is to maintain a consistent temperature variation, while ensuring that TJ, m do not increase. This is particularly important because an increase in its value can significantly reduce the device's lifetime. Although the proposed controller derates the SST with switch degradation, this extended lifetime is significant for maintenance scheduling and avoidance of unexpected failure.

The proposed fast behavioral cascode GaN-FET switch-loss model shows an average mean-squared error of 2.3% and does not require proprietary switch information. The model parameters are easily extractable from the datasheet and DPT, and the loss model does not require exhaustive computation. Though, it is developed for SST, it should be applicable to other power electronics system as well. The integration of health-monitoring systems and degradation-sensitive controllers into digital twin and cloud computing platforms could be useful for industrial and electric-transport maintenance and asset management.

  1. To explore Comparative results with existing approaches/methods relating to the proposed work. The method/approach in the context of the proposed work should be written in detail.

à Answer:

Thank you for your question. We have included three references in our study that explore comparative results with existing approaches/methods relating to the proposed work. We have discussed these references in detail in the introduction section (Introduction Page 2 line 50-75) of our manuscript.

The first reference https://ieeexplore.ieee.org/abstract/document/8999526 [7], discusses the integration of renewable energy sources and new technologies into power systems, which has led to the development of a new reliability framework for modern power electronic systems.

The second reference https://ieeexplore.ieee.org/document/9042353 [8], proposes a comprehensive approach that uses a photovoltaic inverter as an example to predict power electronic converter reliability, address wear-out failures, and optimize power system design, operation, and maintenance.

Finally, in reference https://ieeexplore.ieee.org/document/8718555 [15], the authors propose a power sharing control strategy to improve the reliability of power converters in DC microgrids by adjusting their loadings based on prior thermal damages.

We hope that these additions will address your concerns and improve the quality of our manuscript.

Thank you again for your valuable feedback, and please let us know if you have any further suggestions or comments.

Contents before change (Introduction Page 1-2 line 42-52)

State-of-the-art DAB-control-based protection systems focus on post-fault scenarios [4-8], mainly by carrying out fault tolerance operations after a switch failure. Therefore, they require costly hardware redundancies and high system complexity, which commonly increase the system’s size and weight [4, 5]. Redundancy-based network-level power routing is not feasible in standalone SST-based EV charging applications [7]. In these applications, topology transformation methods can bypass fatal switch failure effects for a limited time. However, these methods double the current through the switch, resulting in higher switch stress and accelerated aging, which causes premature SST failure. They also require computationally exhaustive fault detection and isolation procedures. In [6] and [8], an adaptive but complicated and costly cooling method for junction temperature control was proposed to extend switch life.

Contents after change (Introduction Page 2 line 50-75)

The integration of renewable energy sources and new technologies into power systems has led to the development of a new reliability framework for modern power electronic systems [7]. Another study introduced a comprehensive approach that uses a photovoltaic inverter as an example to predict power electronic converter reliability, address wear-out failures, and optimize power system design, operation, and maintenance [8]. With the increasing demand for more electrical systems, reliability assessment and standardization have become increasingly important [7-8]. The development of guidelines is vital to achieving this objective, and current efforts are focused on creating them. Furthermore, this approach is applicable not only to the entire power electronic system but also to individual components. State-of-the-art DAB-control-based protection systems focus on post-fault scenarios [9-13], mainly by carrying out fault tolerance operations after a switch failure. Therefore, they require costly hardware redundancies and high system complexity, which commonly increase the system’s size and weight [9, 10]. Redundancy-based network-level power routing is not feasible in standalone SST-based EV charging applications [12]. In these applications, topology transformation methods can bypass fatal switch failure effects for a limited time. However, these methods double the current through the switch, resulting in higher switch stress and accelerated aging, which causes premature SST failure. They also require computationally exhaustive fault detection and isolation procedures. A power sharing control strategy is proposed to improve the reliability of power converters in DC microgrids by adjusting their loadings based on prior thermal damages, aiming to extend their lifespan and enhance the overall system reliability [15]. The previous studies [12-15] relied solely on thermal loading information to determine the stage of degradation. In contrast, accelerated life testing (ALT) can provide additional insights into the degradation pattern, which have not been utilized in this study. In [11] and [13], an adaptive but complicated and costly cooling method for junction temperature control was proposed to extend switch life.

  1. At Line 296, what is WBLO?

à Answer:

Thank you for pointing out the abbreviation. We checked and modified it.

Contents before change

WBLO (Section 4 Page 8 line 296)

Contents after change

Wire bond lift-off (WBLO) (Section 4 Page 8 line 321)

  1. In Figure 5. (a), how to determine three stages?

à Answer:

Thank you for your valuable feedback on our manuscript.

In Figure 5(a), the three stages referred to are based on the percentage change in the device's on-resistance (ΔRDS,ON) over time. For more information on this analysis, please refer to Section 7.2 (page 14, lines 467-469) of our manuscript where we provide further details. This was determined from our observation. Based on our observations, we identified three distinct stages of degradation:

  1. Healthy region: This stage is characterized by a ΔRDS,ON of between 0-2%. At this stage, the device is performing well, and there is little to no degradation.
  2. Slow degradation: At this stage, the ΔRDS,ON is between 2-7%. The device is still functioning, but there is some degradation in performance.
  3. Exponential degradation: This is the final stage, where the ΔRDS,ON is greater than 7%. At this point, the device is significantly degraded, and its performance is greatly affected.

  1. The inspiration of your work must be highlighted. For example,

http://dx.doi.org/10.1109/TCSS.2022.3152091;

https://doi.org/10.1111/iwj.13723;

https://doi.org/10.1016/j.ins.2022.12.068;

http://dx.doi.org/10.1016/j.oceaneng.2022.113424 and so on.

à Answer:

Thank you for taking the time to review our paper. We appreciate your feedback and insights. However, we would like to clarify that our study is not related to the DOI examples you mentioned in your review.

Thank you again for your valuable feedback, and we look forward to your response.

http://dx.doi.org/10.1109/TCSS.2022.3152091

A Clinical-Oriented Non-Severe Depression Diagnosis Method Based on Cognitive Behavior of Emotional Conflict

https://doi.org/10.1111/iwj.13723

The amputation and mortality of inpatients with diabetic foot ulceration in the COVID-19 pandemic and postpandemic era: A machine learning study

https://doi.org/10.1016/j.ins.2022.12.068

ABC-GSPBFT: PBFT with grouping score mechanism and optimized consensus process for flight operation data-sharing

http://dx.doi.org/10.1016/j.oceaneng.2022.113424

Experimental and numerical investigation on self-propulsion performance of polar merchant ship in brash ice channel

Reviewer 3 Report

The paper deals with the control of the solid-state transformers. Generally, the paper is well written, both from technical and linguistic point of view. However, there are some issues that need to be addressed, in my opinion.

1.     You say in the Abstract ‘In this paper, a novel, degradation-sensitive, adaptive SST controller for cascode GaN-FETs is proposed. … A fast-behavioral switch-loss model for cascode GaN-FETs is also proposed.’ I consider that you need to express better the main  focus of your study.

2.     The 1st sentence of Section 2 ‘The Coffin–Manson method estimates an application’s remaining useful life based on its operating conditions’ needs a reference.

3.     In subsection 7.2 you say ‘To monitor RDS,ON in real time, VDS,ON was measured using the signal-conditioning circuit described in [12]’. I consider that you need to reformulate as is not supposed that the reader has read all your papers. I think you can find an expression of making a reference to the above-mentioned circuit in another manner.

Author Response

  1. You say in the Abstract ‘In this paper, a novel, degradation-sensitive, adaptive SST controller for cascode GaN-FETs is proposed. … A fast-behavioral switch-loss model for cascode GaN-FETs is also proposed.’ I consider that you need to express better the main focus of your study.

à Answer:

Thank you for your comment on our manuscript. We agree that our abstract should better emphasize the main focus of our study. While we did propose a fast-behavioral switch-loss model for cascode GaN-FETs, the primary contribution of our work is the development of a novel, degradation-sensitive, adaptive SST controller for these devices. We have now revised our abstract to reflect this more clearly. We believe that this revision will better convey the importance and originality of our proposed SST controller, and we thank you for bringing this to our attention.

Contents before change (Abstract Page 1 line 14-26)

Abstract: Solid-state transformers (SSTs) are promising solutions in electric vehicle charging, smart homes, and smart grid applications. Unlike in traditional transformers, a semiconductor switch’s degradation and failure can compromise its robustness and integrity. It is vital to continuously monitor a switch's health condition to adapt it to mission-critical applications. State-of-the-art degradation monitoring methods are computationally intensive, have a limited capacity to identify the severity of degradation, and are challenging to implement in real time. In this paper, a novel, degradation-sensitive, adaptive SST controller for cascode GaN-FETs is proposed. The proposed controller uses accelerated life testing (ALT)-based switch degradation mapping for degradation severity assessment. This controller intelligently derates the SST to 1) ensure robust operation over the SST’s lifetime and 2) achieve the optimal degradation-sensitive function. A fast-behavioral switch-loss model for cascode GaN-FETs is also proposed. This proposed fast model estimates the loss accurately, without proprietary switch parasitic information. The proposed method is validated through a 5kW cascode GaN-FET-based SST platform.

Contents after change (Abstract Page 1 line 14-27)

Abstract: This paper proposes a novel, degradation-sensitive, adaptive SST controller for cascode GaN-FETs. Unlike in traditional transformers, a semiconductor switch’s degradation and failure can compromise its robustness and integrity. It is vital to continuously monitor a switch's health condition to adapt it to mission-critical applications. The current state-of-the-art degradation monitoring methods for power electronics systems are computationally intensive, have limited capacity to accurately identify the severity of degradation, and can be challenging to implement in real-time. These methods primarily focus on conducting accelerated life testing (ALT) of individual switches and are not typically implemented for online monitoring. The proposed controller uses accelerated life testing (ALT)-based switch degradation mapping for degradation severity assessment. This controller intelligently derates the SST to 1) ensure robust operation over the SST’s lifetime and 2) achieve the optimal degradation-sensitive function. Additionally, a fast-behavioral switch-loss model for cascode GaN-FETs is used. This proposed fast model estimates the loss accurately, without proprietary switch parasitic information. Finally, the proposed method is experimentally validated using a 5kW cascode GaN-FET-based SST platform.

  1. The 1stsentence of Section 2 ‘The Coffin–Manson method estimates an application’s remaining useful life based on its operating conditions’ needs a reference.

à Answer:

Thank you for taking the time to review our manuscript and providing valuable feedback regarding the need for references about the Coffin-Manson method.

We agree that including relevant references can help to support and strengthen our arguments. In response to your suggestion, we have added the following reference to the sentence “The Coffin-Manson method estimates an application’s remaining useful life based on its operating conditions”:

[16] Jacques, S., Caldeira, A., Batut, N., Schellmanns, A., Leroy, R., & Gonthier, L. (2011, August). A Coffin-Manson model to predict the TRIAC solder joints fatigue during power cycling. In Proceedings of the 2011 14th European Conference on Power Electronics and Applications (pp. 1-8). IEEE.

The Coffin-Manson method is a well-known technique in the field of reliability engineering that is used to estimate the remaining useful life of an application based on its operating conditions. The method is particularly useful for predicting the fatigue life of mechanical and electronic components that undergo cyclic loading or thermal stress during operation.

Contents before change (Section 2 Page 2 line 84-85)

The Coffin–Manson method estimates an application’s remaining useful life based on its operating conditions.

Contents after change (Section 2 Page 2 line 108-109)

The Coffin–Manson method estimates an application’s remaining useful life based on its operating conditions [16].

  1. In subsection 7.2 you say ‘To monitor RDS,ON in real time, VDS,ON was measured using the signal-conditioning circuit described in [12]’. I consider that you need to reformulate as is not supposed that the reader has read all your papers. I think you can find an expression of making a reference to the above-mentioned circuit in another manner.

à Answer:

Thank you for bringing this to our attention. We appreciate your feedback and have taken it into consideration. After careful consideration, we have rephrased the sentence (Subsection 7.2 Page 13 line 433-434) to better convey the information to the reader:

Contents before change (Subsection 7.2 Page 13 line 433-434)

To monitor RDS,ON in real time, VDS,ON was measured using the signal-conditioning circuit described in [12]

Contents after change (Subsection 7.2 Page 13 line 458-460)

To measure VDS,ON and monitor RDS,ON in real-time, a signal-conditioning circuit was utilized. For a detailed description of this circuit, please refer to our previous publication [19]

Round 2

Reviewer 1 Report

In my opinion, now this paper may be published.

Reviewer 3 Report

Thank you for answering to my comments.